# The Effects of Microbiota on the Herbivory Resistance of the Giant Duckweed Are Plant Genotype-Dependent

**DOI:** 10.3390/plants11233317

**Published:** 2022-12-01

**Authors:** Martin Schäfer, Shuqing Xu

**Affiliations:** 1Institute for Evolution and Biodiversity, University of Münster, 48149 Münster, Germany; 2Institute of Organismic and Molecular Evolution, Johannes Gutenberg University Mainz, 55128 Mainz, Germany

**Keywords:** *Spirodela polyrhiza*, giant duckweed, *Lymnaea stagnalis*, great pond snail, microbiota, evolution, adaptation, herbivory, tolerance, resistance

## Abstract

In nature, all plants live with microbes, which can directly affect their host plants’ physiology and metabolism, as well as their interacting partners, such as herbivores. However, to what extent the microbiota shapes the adaptive evolution to herbivory is unclear. To address this challenge, it is essential to quantify the intra-specific variations of microbiota effects on plant fitness. Here, we quantified the fitness effects of microbiota on the growth, tolerance, and resistance to herbivory among six genotypes of the giant duckweed, *Spirodela polyrhiza.* We found that the plant genotypes differed in their intrinsic growth rate and tolerance, but not in their resistance to a native herbivore, the great pond snail. Inoculation with microbiota associated with *S. polyrhiza* growing outdoors reduced the growth rate and tolerance in all genotypes. Additionally, the microbiota treatment altered the herbivory resistance in a genotype-specific manner. Together, these data show the potential of microbiota in shaping the adaptive evolution of plants.

## 1. Introduction

All plants in nature live with microbes that populate their close proximity, live on their surface, or even live directly within them [1,2]. As plant microbiota can profoundly change the metabolism, physiology, and fitness of the host plants, the intimate interaction between plants and their microbiota can also affect their interaction with herbivores [3] and thus potentially also the evolutionary trajectory of adaptation to herbivory. While increasing studies suggest that the plant microbiota alters plant growth and defenses [3,4,5], they have largely focused on single genotypes. However, to understand whether and how the plant microbiota affects plant evolution, it is essential to quantify the effect of microbiota on plant fitness using different plant genotypes (fitness landscape).

Here, we address this challenge using the hydrophyte, *Spirodela polyrhiza* (Araceae, Lemnoideae), one of the fastest-growing angiosperms, in which the microbiota can be manipulated to assess the microbe-dependent plant fitness effects [6,7]. For example, inoculation of *S. polyrhiza* with *Ensifer sp*. strain SP4 can promote plant growth [8], and *S. polyrhiza*-associated microorganisms can contribute to the degradation of microcystins, which are a group of cyanotoxins [9]. Using six globally distributed *S. polyrhiza* genotypes, we showed that the microbiota reduced plant growth and tolerance to herbivory by freshwater snails, while it changed the resistance to herbivory in a genotype-specific manner, indicating the importance of microbiota in shaping the evolutionary trajectory of plants in nature.

## 2. Results

### 2.1. Different Genotypes Varied in Intrinsic Growth Rate, but Not Resistance to Herbivory by Snails

To investigate the genotype-dependent fitness in *S. polyrhiza*, we first analyzed the growth rate of six different *S. polyrhiza* genotypes originating from Asia, North America, Australia, and Europe. We calculated the relative growth rate (RGR) based on the changes in frond numbers within 7 days. As each frond represents an individual plant, the frond number is strongly correlated with fitness in *S. polyrhiza*, although additional measurements of biomass, surface area, and chlorophyll content could further improve our mechanistic understanding. Under the greenhouse conditions, the six genotypes showed different RGR (Figure 1A, genotype effects: *p* = 2.2 × 10^−16^, Kruskal–Wallis rank sum test). While genotype SP05 showed the highest growth rate, genotype SP30 had the lowest growth rate.

We then performed bioassays to quantify the tolerance after snail herbivory, based on the RGR within 5 days after snail exposure (RGR_tolerance_). During snail exposure, snails were allowed to feed on the duckweed populations for 24 h. The RGR_tolerance_ was calculated using the remaining fronds after snail exposure, which still possessed at least one of the reproductive pouches (including damaged fronds) and therefore were expected to still have the potential to further reproduce, as a starting point. The six genotypes showed different RGR_tolerance_ (Figure 1B, genotype effects: *p* = 0.031, Kruskal–Wallis rank sum test). While genotype SP14 showed the highest RGR_tolerance_, genotype SP02 had the lowest RGR_tolerance_.

Additionally, we quantified the plant resistance to snail herbivory based on the number of consumed or damaged fronds by one snail within 24 h. High levels of plant damage would represent a low plant resistance to herbivory. On average, each snail consumed approximately 20–25 fronds. However, no statistical difference was found among the genotypes (Figure 1C, genotype effects: *p* > 0.1, one-way ANOVA).

### 2.2. Microbiota Inoculation Altered Tolerance and Resistance to Herbivory in a Genotype-Specific Manner

To examine whether the microbiota affects the growth rate, tolerance, and herbivory resistance, we inoculated six genotypes with microbiota collected from a natural population of *S. polyrhiza*. To prevent algae overgrowth, the inoculum was filtered through a 3 µm filter, which is expected to also remove most fungal cells. Therefore, the inoculum should mainly consist of the bacteria associated with duckweed and its environment. It still might have included some microalgae such as the prokaryotic cyanobacteria, but no excessive algae growth was visually observed. Interestingly, inoculation of microbiota reduced the growth rate (Figure 1D, *p* = 1.8 × 10^−6^) and tolerance (Figure 1E, *p* = 6.4 × 10^−5^) of all genotypes, but no statistical differences were found among the genotypes (*p* > 0.1).

We then quantified the effects of the microbiota on herbivore resistance. Among the six genotypes, the effects of microbiota on resistance differed (Figure 1F, *p* = 0.069). The microbiota inoculation decreased the resistance of genotype SP05 to herbivory by 23%, while it had an opposite effect on genotype SP14 by increasing its resistance by 41% (*p* = 0.014).

## 3. Discussion

In nature, plants interact with complex biotic interaction partners, such as beneficial or detrimental microbes, and herbivores. The evolutionary trajectory of a plant therefore not only depends on one interaction partner, but the combination of different interacting members in the community. Consistent to this prediction, we found that the interaction of both an herbivore and the microbiota might alter the fitness landscape among different *S. polyrhiza* genotypes.

We found different *S. polyrhiza* genotypes differed in their intrinsic growth rate and tolerance. Interestingly, microbiota inoculation reduced the growth rate and tolerance in all genotypes, and no significant differences were found among the genotypes. Many previous studies reported the microbe-induced growth promotion of duckweed [8,10,11], but also negative growth effects have been reported before [11,12]. Overall, we observed a relatively low growth rate, which could be caused by non-optimal growth conditions in the greenhouse and the comparably low nutrient concentration of the medium. However, in nature, the growth conditions for *S. polyrhiza* are also often suboptimal and dependent on the season, the geographic location, and the surrounding vegetation, while nutrient levels vary between different water bodies and can also change over time [13].

Mechanistically, the observed effects of the microbiota on plant growth, tolerance, and resistance could be caused by at least three different non-exclusive mechanisms. First, the microbiota might have induced changes in signaling pathways either via microbial elicitors, such as flagellin, or via microbe-produced bioactive compounds, such as phytohormones [11,14], that could directly affect the plants physiology and metabolisms and/or its defense response to herbivory. Second, the microbiota might have altered the nutrient supply of the *S. polyrhiza* plants. For example, some duckweed-associated microbes can solubilize insoluble phosphate, produce siderophores, or improve nitrogen uptake [8,11]. In contrast, other microorganisms, such as algae can compete with duckweed for the available nutrients in the medium [12]. The nutrient availability can further affect the plants primary and secondary metabolisms [15], which are key components of plant defenses. Third, the microbiota might have produced specific toxins that directly affected the plant or deterred the feeding from the snails. For example, the endophytic fungus, *Acremonium coenophialu*, in tall fescue (*Festuca arundinace*) produces alkaloids that deter herbivores on its host [16] and *Lemna minor*-associated microbes are able to produce hydrogen cyanide that can inhibit the plant growth [11]. Furthermore, the genotypes might assemble different microbe communities despite the treatment with the same inoculum [17] and therefore differentially affect the plants. Although most of the available literature in this field is based on terrestrial plants, the conserved structure of the duckweed microbiome and the terrestrial leaf microbiome indicate similar mechanisms underlying the assembly of the associated microbe community [18]. In the future, the combination of multi-omics profiling, synthetic microbe communities, and genetic manipulation approaches in *S. polyrhiza* will shed light on the mechanisms involved in plant–microbe–herbivore interactions.

A major limitation in studying plant–microbe–herbivore interactions is the efficiency and stability of microbiota inoculation. The microbiota consists of different living microbes that can dynamically change their composition depending on the experimental conditions, such as the pH, temperature, and chemical composition of the media used for extraction, inoculation, and plant cultivation. Therefore, we currently do not know to what extent the inoculation of microbiota reassembled the natural microbiota in our experiments. Differences between a field microbiome and the respective inoculum have been found before [10], although fewer differences might be expected from protocols without a microbe cultivation step, e.g., by direct inoculation with freshly isolated microbes. Additionally, during growth under semi-sterile conditions, the control plants might also assemble a specific microbiome and during some steps, such as the addition of the non-sterile snails, further changes in the microbiome can occur. Sequencing the microbiota using meta-barcoding might offer a glimpse into the differences between natural and re-inoculated microbiota. Additionally, a more natural way of inoculation is to grow sterile genotypes together with outdoor growing *S. polyrhiza* for several generations. Consistently, we also found genotype-dependent microbiota effects on plant resistance in a current study using such a natural inoculation approach in a long-term experimental evolution experiment [19].

In summary, this study demonstrates the potential of microbiota in shaping the evolutionary trajectory of plants in nature, highlighting the need to consider microbiota in studying plant evolution and plant–herbivore interactions.

## 4. Methods

### 4.1. Duckweed Growth with and without Microbiota-Inoculation

*Spirodela polyrhiza* plants of the genotypes SP1 (ID 0040, China), SP2 (ID 6613, USA), SP5 (ID 7551, Australia), SP9 (ID 8756, Ethiopia), SP14 (ID 9509, Germany), and SP30 (ID 8442, India) were pre-cultivated in Erlenmeyer flasks under sterile conditions for 3 weeks with high nutrient availability in N-medium (KH_2_PO_4_ 150 μM, KNO_3_ 8 mM, Ca(NO_3_)_2_ 1 mM, H_3_BO_3_ 5 μM, MnCl_2_ 13 μM, Na_2_MoO_4_ 0.4 μM, MgSO_4_ 1 mM, FeNaEDTA 25 μM) [20] to generate sufficient fronds. Subsequently plants were transferred to Hoagland medium (NaH_2_PO_4_ 32 μM, KNO_3_ 357 μM, H_3_BO_3_ 15 μM, MnCl_2_ 1.97 μM, H_2_MoO_4_ 1.09 μM, MgSO_4_ 418 μM, K_2_SO_4_ 1.68 mM, CuSO_4_ 120 nM, ZnSO_4_ 278 nM, FeSO_4_ 12 μM, CaCl_2_ 1 mM; pH 7.0) [11] with reduced nutrient availability to acclimate for 3 days before the microbe inoculum or a buffer control was added. The plants were then incubated for 5 days in the medium supplemented with the microbe inoculum. Inoculation was conducted for all plants of each genotype together and all of the genotypes were treated with a subfraction of the same microbe isolate. At the beginning of the experiment (t0) for each replicate, 20 fronds were transferred under semi-sterile conditions to plastic cups (PP, transparent, round, 250 mL, Plastikbecher.de GmbH) containing 150 mL of fresh Hoagland medium. The plastic cups were closed with a perforated lid (PP, transparent, round, 101 mm, Plastikbecher.de GmbH). The plants originate from our in-house stock collection, which is kept at 18 °C, with a day/night period of 12:12. With the start of the pre-cultivation step, the plants were grown under glasshouse conditions between August and September 2018 without light supplementation or temperature regulation. Their growth was measured after 7 days (t7), by counting the frond number (FN), to calculate the relative growth rate with the formula RGR = (LnFN_t7_ − LnFN_t0_)/(t7 − t0) [21]. An overview of the experimental procedure is shown in Figure 2. The RGR values were largely affected by the direct offspring of the starting fronds. Genotype-dependent microbiota effects were calculated based on the relative difference between the plants treated with and without microbe inoculum.

### 4.2. Microbe Inoculum

Microbes were extracted in a similar way to the method as described by Ishizawa et al. (2017) [11]. In brief, *S. polyrhiza* plants from a population that was under herbivory pressure outdoors and water from these ponds were used as starting materials for the microbe inoculum. An approximately similar number of fronds was used as the number of fronds supposed to be treated. Duckweed plants were homogenized in 5 mg/L sodium tripolyphosphate (Supelco) using a kitchen blender. Subsequently, the water and plant homogenate were filtered with a coffee filter and then through a 3.0 µm filter (MCE membrane, MF-Millipore, Merck, Germany). The flow-through was centrifuged (10 min, 10,000× g, 4 °C) to collect the microbes. The pellet was washed twice with Hoagland medium and subsequently re-suspended in Hoagland medium. The resulting microbe suspension was used for inoculation and pure Hoagland medium as a buffer control.

### 4.3. Herbivore Assay

At the end of the growth assay (t7), one pond snail, *Lymnaea stagnalis*, was added per cup and the number of remaining fronds was counted after 24 h (t8). The snails were removed and 5 days later (t13), the fronds were counted again. Tolerance was defined as the RGR observed after the snail feeding RGR_tolerance_ = (LnFN_t13_ − LnFN_t8_)/(t13 − t8) including intact and damaged fronds. Herbivore resistance was determined based on the number of fronds consumed or damaged by the snail, which was calculated as the difference between the observed undamaged fronds and the number of expected fronds for day 8 (FN_t8expected_–FN_t8_). FN_t8expected_ was calculated based on the frond number on day 7 and the RGR between t0 and t7. An overview of the experimental procedure is shown in Figure 2B. The snails were collected from an outdoor experimental pond that was covered by duckweed (*L. minor*), cleaned with tap water, and starved for one day before the start of the experiment. The snails used for the experiment were sorted by size and the snails with a length of approximately 2 cm were used. The snails were randomly added to the cups.

### 4.4. Statistics and Data Analysis

All data were analyzed in R v4.2.1 [22]. Differences in growth rate and resistance among genotypes were tested using linear models. Non-parametric tests were used when the data were not normally distributed. The microbiota effects were respectively normalized to the trait of each genotype. The original data and data analysis R-scripts are provided in a Appendix A. The cups that were used for the growth, herbivore, and tolerance assay were treated as experimental units and the number of replicates is provided in the figure legend. For the growth assay, the genotypes SP1, SP2, SP5, SP14 and SP30 had 20 replicates, and SP9 had 16 replicates. For the tolerance and herbivore assay, SP1, SP2, SP5, SP14 and SP30 had 15 replicates, and SP9 had 11 replicates.

## Figures and Tables

**Figure 1 plants-11-03317-f001:**
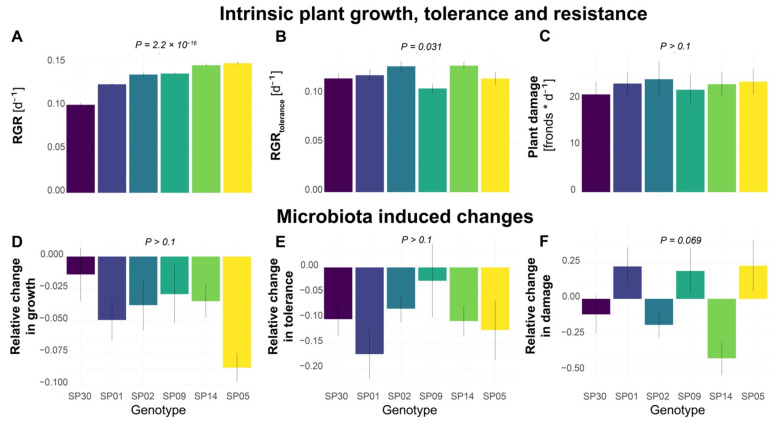
Intra-specific variations of microbiota effects on plant growth, tolerance, and resistance to the snail herbivory in *S. polyrhiza.* (**A**) Relative growth rate (RGR) based on the number of fronds grown within a week. (**B**) Tolerance as RGR within 5 days after snail herbivory (RGR_tolerance_). (**C**) Plant damage, which was calculated as the number of fronds consumed or damaged by one *L. stagnalis* within 24 h and represents the level of plant resistance to herbivory, with high damage levels indicating low resistance. (**D**–**F**) The effects of plant microbiota on growth rate, tolerance, and resistance to snail herbivory, estimated as the percentage of the growth/tolerance/damage rate that was altered by the pre-treatment with a microbe inoculum. Positive and negative values refer to increased or decreased growth/tolerance/resistance levels. Columns represent mean values and error bars show the standard error. Each column represents a genotype with N > 16 for (**A**,**D**) and N > 11 for (**B**,**C**,**E**,**F**). *p*-values refer to genotype effects.

**Figure 2 plants-11-03317-f002:**
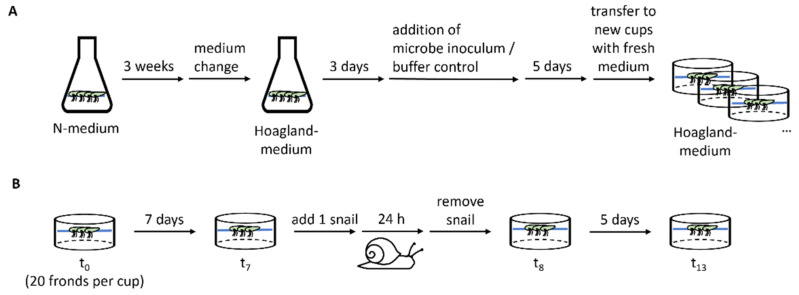
Overview of the experimental procedure. (**A**) Scheme of the plant pre-treatment, including plant propagation on N-medium, adaptation to Hoagland medium, microbe inoculation, and incubation before the plants were divided to the cups for conducting the experiment. For each genotype, one Erlenmeyer flask was used per treatment. (**B**) Experimental procedure to determine the plant growth before (t_0_–t_7_) and after herbivory attack (t_8_–t_13_) as well as the herbivore damage within 24 h.

## Data Availability

The data presented in this study are available in the Appendix A.

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
