# Peer review of "The Effects of Microbiota on the Herbivory Resistance of the Giant Duckweed Are Plant Genotype-Dependent"

_plants, 2022, doi:10.3390/plants11233317_

Round 1
Reviewer 1 Report
This manuscript briefly summarized their experiment on plant-microbe-herbivore interactions. Although the study seems somewhat preliminary and some drawbacks in experimental design exist, the obtained results are new and should serve as a starting point for further interesting works by the authors. My comments below are largely editorial and do not question the overall value of this manuscript.
1. While I appreciate the value of this manuscript as a starting point to deepen our understanding of plant adaptive evolution in nature, the strong statements in the context of evolutionary biology (e.g., the last sentence of Introduction and the last paragraph of Discussion) seems not to match the current experimental results. In particular, I afraid that the term “fitness landscape” in the title promise the readers more than what the manuscript covers.
2. It is likely that different duckweed genotypes assembled different microbiota composition even from the same microbial inoculum. Furthermore, the control plants that were cultured in “semi-sterile” conditions might assembled specific microbiota around them, depending on the degree of contaminations. These facts may affect the author's experimental design and interpretation and are expected to be discussed in this manuscript.
3. Please describe more experimental details on snail exposure at the second paragraph of the Results.
4. The same scales of y-axes in Fig 1A and Fig1B, Fig 1D and Fig. 1E are more preferred.
5. In Fig. 1C and 1F, high “resistance” indicates larger damage by herbivore? This seems somewhat complicated.
6. The authors seemed to focus on plant-bacterial interactions, since the 3.0 µm filter should have removed the most fungal cells associated to natural duckweeds. This should be made explicit.
7. Please clarify how did you obtain the snails and control their conditions (e.g., age and size) for experiments.
8. Please clarify the approximate range of temperature, light intensity and photoperiod (if possible) in the greenhouse.
Reviewer 2 Report
Title: Microbiota alter the herbivory-dependent fitness landscape in the giant duckweed
The manuscript investigated the impact of natural microbe community on the growth of several genetically different strains of Spirodela polyrhiza. The other aim of the study was how microbial community alter resistance of these plant against snail herbivory and trough this how they shape plant fittness. The idea is novel and add some initial info to plant adaptation and how microbiota shapes evolutionary trajectory in natural systems.
I have some critical points for improving the manuscript in case of acceptance.
1. Introduction:
“”using the diploid hydrophyte, Spirodela” : “diploid” should be skipped since it is self evident that all Angiosperms are diploids.
The sentence is misleading: “it is essential to quantify genotype-specific effect of microbiota on plant fitness”
It indicates that authors used different genotypes of microbae instead of plant starins.
I would use: it is essential to quantify effect of microbiota on plant fitness using various plant genotypes.
Actually I would avoid using the term fitness in the MS, since simply the number of the fronds was counted and growth rate of the plants was calculated. My opinion is that if growth rate is calculated only by number based on the short term experiment, it can not be make equal to fitness.
4. Methods
The text should be crystal clear for the readers including each and every step of the method. Brief flow-chart should be useful.
What was the reason that plants were preincubated on two different media: N medium (for 3 weeks) and then placed to Hogland-medium for 3 days.
What is the nutrient concentration (N, P) of the used media?
This sentence should be more strait: “subsequently plants were transferred to Hoagland-medium to acclimate for 3 days, before the microbe inoculum or a buffer control was added. Plants were then inoculated for 5 days”.
Does it mean that you inoculated the plants for five days?
I think plants was inoculated and then incubated for 5 days.
Based on the main text in the MS it is remained hidden how many experimental units (Erlenmayer flask) were used for the experiment? The number of replicates is also not shown in figure caption. The meaning of error bars on Fig 1 should be indicated.
It is pity that frond number as the only single plant trait was measured for evaluating the impact of microbae on plant growth. This trait is loaded with quite high subjective error (see Landolt and Kandeller 1987). In the 21th century, fresh biomass, dry mass, leaf area and especially chlorophyll content would indicate more sensitively the impact of the treatments (microbae).
Indicating the range of light and temperature conditions in the green house would be useful for the readers.
Based on the RGR data, the growth are quite low (RGR< 0.15 day-1) even in axenic plants cultures comparing to other duckweed assays. This indicated that growth conditions of the plants were not in the optimal range. What was the daily day/night photoperiod for the axenic strains under the period they were cultivated in axenic cultures before the experiment?
Conducting an experiment in greenhouse condition at the end of the growing season (September-October) is not the best choice because the growth is decreasing since lowering the day light hours.
4.2. Microbe inoculum
Using 3 um size of of filter, several microalgal species could also exist into the inoculated microbial community. This can cause algal growth within 2-4 days resulting high pH and low nutrient concentration in the medium competing with duckweeds (see relevant literatures).
Beyond competition for nutrients, inhibitory effects of physical contact of algae or other microbae can be another type of negative effect on duckweed growth (see relevant literatures).
2. Results:
“We then quantified the effects of the microbiota on herbivore resistance using a 24- hour herbivory bioassay.” The sentence should be skipped since it was mentioned in Methods.
Among six genotypes, replaced to Among the six genotypes,
“For example,” should be deleted.
Discussion:
The discussion section is full of vague assumptions without any experimental data:
“the microbiota might have induced changes in signaling pathways either via microbial elicitors, such as flagellin or via microbe-produced bioactive compounds, such as phytohormones, that could directly affect the plants physiology and metabolisms and/or its defense response to herbivory.”
“Second, the microbiota might have altered the nutrient supply of the S. polyrhiza plants”. No experimental data.
In literature references, comparisons are often based on terrestrial environmental results (grass species like rice and Festuca).
Vague sentence: “The nutrient availability then can have profound effects on the plant primary and secondary metabolisms”
“Profound effect” does not mean anything in Plant Science, even whether it is negative positive impact.
All together, four literatures (8-10) were involved in the discussion and only one single literature was about Spirodela. I would highly advice collecting and comparing those basic literatures dealing with the impact of microbae (heterotrophic bacteria, cyanobacteria, eucaryotic algae, fungi) on duckweed including facilitation, competition, allelopathy.
Round 2
Reviewer 2 Report
Second round submission
The manuscript has improved in several points where there was a chance to improving. However, the following parts are remained without any improvement.
It is still remained hidden how many experimental units (Erlenmayer flask) were used for the experiment? The number of replicates is also not shown in figure caption.
Based on the low RGR data, it seems that experiment was conducted under suboptimal light and temperature conditions. This is one of the weak point of the study.
Neither figure legend, nor Method section indicated the exact number of replicates.
Among the several more precisely measurable plant traits (biomass, total chlorophyll, only frond number was measured. This is the weakest point of the study.
Fig2. There is not a single duckweed frond in the experimental units. Snail is the only living creature is visible here. Please add Spirodela to the experimental units.
Discussion has been improved by adding several examples from aquatic environment. However, it is still not clearly structured. I still see mainly diverse and mixed information with full of assumption.
